# Deep versus Moderate Neuromuscular Blockade in Gynecologic Laparoscopic Operations: Randomized Controlled Trial

**DOI:** 10.3390/jpm12040561

**Published:** 2022-04-01

**Authors:** Nikolaos Kathopoulis, Athanasios Protopapas, Emmanouil Stamatakis, Ioannis Chatzipapas, Dimitrios Zacharakis, Themos Grigoriadis, Stavros Athanasiou, Dimitrios Valsmidis

**Affiliations:** 1Department of Obstetrics and Gynaecology, Alexandra Hospital, National and Kapodistrian University of Athens, 11528 Athens, Greece; prototha@otenet.gr (A.P.); ixatzipapas@yahoo.gr (I.C.); dimzac@hotmail.com (D.Z.); tgregos@yahoo.com (T.G.); stavros.athanasiou@gmail.com (S.A.); 2Department of Anesthesia, Alexandra Hospital, 11528 Athens, Greece; emmstamatakis@hotmail.com (E.S.); dimivals@otenet.gr (D.V.)

**Keywords:** benign gynecology, deep neuromuscular blockade, laparoscopy

## Abstract

Background: To investigate whether deep neuromuscular blockade (NMB) improves surgical conditions and postoperative pain compared to moderate block, in patients undergoing gynecologic laparoscopic surgery. Methods: A single blind, randomized, controlled trial was undertaken with laparoscopic gynecologic surgical patients, who were randomly assigned to one of the following two groups: patients in the first group received deep NMB (PTC 0-1) and in the other, moderate NMB (TOF 0-1). Primary outcomes included assessing the surgical conditions using a four-grade scale, ranging from 0 (extremely poor) to 3 (optimal), and patients’ postoperative pain was evaluated with a five-grade Likert scale and the analgesic consumption. Results: 144 patients were analyzed as follows: 73 patients received deep NMB and 71 moderate NMB. Mean surgical field scores were comparable between the two groups (2.44 for moderate vs. 2.68 for deep NMB). Regarding postoperative pain scores, the patients in the deep NMB experienced significantly less pain than in the group of moderate NMB (0.79 vs. 1.58, *p* < 0.001). Moreover, when the consumption of analgesic drugs was compared, the moderate NMB group needed more extra opioid analgesia than the deep NMB group (18.3% vs. 4.1%, *p* = 0.007). From the secondary endpoints, an interesting finding of the study was that patients on deep NMB had significantly fewer incidents of subcutaneous emphysema. Conclusions: Our data show that, during the performance of gynecologic laparoscopic surgery, deep NMB offers no advantage of operating filed conditions compared with moderate NMB. Patients may benefit from the deep block as it may reduce postoperative pain.

## 1. Introduction

Over the last three decades, laparoscopy has been gradually expanding its use in gynecological surgery and constitutes, today, a valid alternative to laparotomy. Improvements in laparoscopic instrumentation, energy sources, and techniques have enabled surgeons to increasingly utilize this modality in a large number of gynecological procedures [1,2]. Reduced intraoperative blood loss, less postoperative pain, and need for analgesia, faster recovery, shorter hospitalization times, and the better cosmetic outcome are some of the recognized advantages of laparoscopy [3]. One of the essential prerequisites for an effective laparoscopic operation is continuously maintaining an excellent operative field throughout the procedure [4].

With its different aspects, anesthesia plays an important role, affecting the outcome of any procedure. With its inherent characteristics of a limited operative field and a constant need for precise manipulations, laparoscopy is strongly dependent on anesthesia. Neuromuscular Blockade (NMB) and the degree of its depth may affect the surgical conditions [5]. In theory, deep NMB could provide a better relaxation of the diaphragm and abdominal wall musculature than a moderate block, increasing the space between the abdominal wall and the intrabdominal organs during laparoscopy [6]. Studies have shown that deep NMB may result in an improved quality of surgical conditions in laparoscopic prostatectomy and laparoscopic cholecystectomy [7,8]. Moreover, the combination of deep NMB with lower insufflation pressure reduces postoperative shoulder pain, which is often a side effect of laparoscopic surgery [9].

This randomized controlled study aimed to examine if deep NMB may improve operating field visualization and reduce postoperative pain, in patients undergoing gynecologic laparoscopic operations compared to moderate NMB. So far, data regarding the effect of deep NMB on gynecologic laparoscopy are controversial, and our study could contribute to understand if this type of anesthesia may have any positive clinical impact on the patient.

## 2. Materials and Methods

### 2.1. Study Design and Patients

This is a randomized controlled trial performed between September 2017 and November 2019 at the 1st Department of Obstetrics and Gynecology of the University of Athens (Athens, Greece). Female patients, over 18 years old, ASA I-II, with the indication of laparoscopic gynecologic operation were approached at least two weeks before the surgery. Exclusion criteria included insufficient knowledge of the Greek language, patients with known gynecological cancer, hepatic or renal failure, and known allergy to sugammadex. The trial was registered in the ISRCTN registry (No 59502124) assigned on 30 August 2017. The study was conducted in accordance with the CONSORT statement guidelines (Appendix A) [10].

### 2.2. Ethics

Research ethics approval was provided by the institutional board of the “Alexandra Hospital” (No 74/12 February 2015), which is a tertiary referral center for Gynecologic Endoscopic Surgery in Greece, and written informed consent was obtained from all patients.

### 2.3. Randomization and Blinding

Patients were randomized, based on a computer-generated list, to deep or moderate NMB during the operation’s whole length. The allocation was concealed in sealed opaque envelopes. The operating surgeons, the patients, and the researchers were blinded to treatment allocation, while the attending anesthetist was not blinded because he was responsible for maintaining the depth of NMB.

### 2.4. Anesthesia

In both groups, the induction in anesthesia achieved after 3 min of 100% oxygen exposure (target EtO2 = 80%) with intravenous propofol (2 mg/kg at induction and 6 mg/kg/h continuously during the surgical procedure), fentanyl 4 μg/kg and morphine 0.15 mg/kg. Both groups received an initial dose of Rocuronium 1.2 mg/Kg and a maintenance dose of 0.25 mg/Kg. In deep NMB, the maintenance dose was given when two post-tetanic responses (PTC) appeared, in moderate NMB when two responses appeared in the Train of Four stimulation (TOF). Neuromuscular function was monitored continuously by NMT-DATEX^®^ kinemyography, after stabilizing the muscle response monitored by peripheral nerve stimulation on the right hand with two stimulating electrodes applied to the wrist along the ulnar nerve. Monitoring of the neuromuscular function continued to the end of the procedure and every 3 min. Sugammadex was administered at the end of surgery to reverse any residual NMB determined by TOF count (4 mg/kg if TOF count was <2 and 2 mg/kg if TOF count was >2). The monitoring included electrocardioscopy, noninvasive blood pressure measurement, intrapulmonary pressures, end-tidal CO_2_ (ΕtCO_2_), and pulse oximetry. All patients received pressure-controlled ventilation-volume guaranteed ventilation (PCV-VG) mode with target ΕtCO2 34–36 mmHg.

### 2.5. Surgery and Evaluation of Surgical Conditions

Laparoscopic surgery was performed using a Veress needle to establish pneumoperitoneum at an initial abdominal pressure of 20 mm Hg. After the trocar placement, the pressure was set to 12 mm Hg, and the patient was positioned to 30° of Trendelenburg. Two 5 mm trocars and one 10/12 mm trocar were subsequently introduced. The operating surgeon was responsible for the visual field scoring during the surgery, and for that reason, he was blinded to the randomization group of the patient involved. Once the camera was installed through the umbilical port, the surgeon in charge assessed the surgical field’s exposure on a four-grade Likert scale, previously used in the literature: excellent 3; good but not optimal 2; poor but acceptable 1; unacceptable 0 [11]. The same grading was used to evaluate the small and the large bowel, which usually interferes with the visual field during gynecologic laparoscopic surgery.

### 2.6. Postoperative Protocol

Postoperative pain was reduced by paracetamol (1000 mg every 8 h) and diclofenac (75 mg every 12 h). In case of insufficient analgesia, morphine 50 mg i.v., was given upon the patient’s request. On a postoperative day, the analgesic was provided per os. The postoperative pain was evaluated with a 5-point Likert scale 24 h after the operation ranging from 0 to 4, where 0 indicates no pain, 1 slightly bothersome, 2 definitely bothersome, 3 very bothersome, and 4 distressing.

### 2.7. Outcome Measures

The two main endpoints were the quality of the surgical field and the postoperative pain. The postoperative pain was evaluated using a Likert scale rating by the patient on the first postoperative day and extra opioid analgesia consumption from the patients. Secondary outcome measures included operation time, gas volume consumption, hemoglobin difference, hospital stay, postoperative bloating and nausea, postoperative bowel function, and presence of intraoperative subcutaneous emphysema.

### 2.8. Statistics

Determining the sample size for the study’s primary outcome, the surgical field’s quality was based on a prior study reporting 80% of the operations to have an acceptable rating (excellent or good) in surgical field visualization [11]. To demonstrate a 10% difference between the treatment groups, with 80% power at the 5% significance level, we calculated 138 women. We randomized 150 women to account for dropout. Data were recorded and analyzed using SPSS version 20.0 software and Minitab 16.0 software. For quantitative demographic data, parametric analysis of variance with Student *t*-test and one-way ANOVA were used, and non-parametric (Kruskal–Wallis test and Mann–Whitney U test) were used where necessary. Post hoc analysis for multiple comparisons was undertaken by the Bonferroni test when variance was equal and Dunnett T3 when unequal variance was demonstrated. Chi-square test was used to test for independence between categorical variables. The level of statistical significance was set at 5%.

## 3. Results

Between September 2017 and April 2019, 195 women were assessed for eligibility, of whom 144 were analyzed. In terms of the groups, there were 71 patients in the moderate NMB group and 73 in the deep NMB group. The study flow diagram is demonstrated in Figure 1.

The two groups were comparable in terms of patients’ demographic characteristics (Table 1). On average, women in the study were 37 years old, with a mean BMI of 24; 20% of patients were regular smokers, 32% had a history of a prior surgery with laparoscopy or laparotomy, and 24% had a history of daily drug therapy. The distribution of performed laparoscopic procedures was as follows: ovarian cystectomy (58 patients, 40%), myomectomy (39 patients, 27%), hysterectomy (26 patients, 18%), bilateral ovarian cystectomy (10 patients, 7%), deep endometriosis node resection (6 patients, 4%) and salpingo-oophorectomy (5 patients, 2%). Operative data also did not differ significantly between groups (Table 1). Mean surgery time from anesthesia introduction to patient extubation was 104 min (range: 60–385 min), gas volume consumption was 541 Lt (SD: ±336 Lt), and mean hemoglobin difference was 0.96 g/dL (SD: ±0.74 g/dL).

Mean surgical field scores were 2.44 for moderate NMB versus 2.68 for deep NMB (0.06). Small and large bowel handling scores were also comparable between the two groups (2.63 vs. 2.79 and 2.25 vs. 2.51, respectively) (Table 2). When a subgroup analysis regarding operation duration was implemented (≤90 min and >90 min), the results did not change. Specifically, the overall operating field and the small and large bowel handling were comparable between the groups of moderate and deep NMB, irrespectively of the operation’s duration (Table 3). Regarding patients’ postoperative pain scores, the patients in the group of deep NMB experienced significantly more pain compared to the group of moderate NMB (0.79 vs. 1.58, *p* <0.001) (Table 4). Moreover, when the consumption of analgesic drugs was compared, the moderate NMB group needed more extra opioid analgesia than the patients in the deep NMB group (18.3% vs. 4.1%, *p* = 0.007) (Table 5).

Postoperatively, patients on the moderate NMB experienced significantly more severe symptoms of nausea (1.13 vs. 0.67, *p* = 0.02) and bloating (1.11 vs. 0.79, *p* = 0.03) compared to patients in the deep NMB group. There was no difference on the day that patients experienced the passage of flatus and stools from the anal verge for the first time after the surgery (Table 4). The hospital stay was also comparable between the two groups (Table 1).

There was a trend towards fewer incidents in patients in the deep NMB group, regarding complications, and specifically for subcutaneous emphysema, although no statistical significance was reached (6.8% vs. 16.9%, *p* = 0.06) (Table 6). The subgroup analysis, according to operation duration, revealed a significant difference in subcutaneous emphysema occurrence in patients receiving deep NMB, whose operations lasted >90 min (10% vs. 28%, *p* = 0.04) (Table 6).

## 4. Discussion

This study shows that deep NMB failed to improve operative field visualization compared to moderate block in gynecologic laparoscopic surgery. Moreover, there was no benefit in small and large bowel handling in patients who received deep NMB. The same results were maintained, even when subgroup analysis regarding operative time was performed. Deep NMB was superior compared to moderate block regarding postoperative pain, and patients consumed fewer analgesics.

Only a few studies examined the effect of deep NMB in gynecologic laparoscopic surgery. Madsen et al. measured the distance from the sacral promontory to the trocar during deep NMB and without NMB at pneumoperitoneum 8 and 12 mmHg and discovered that deep NMB improved surgical space mean 0.3 and 0.33 cm, respectively (*p* < 0.05) [12]. The authors, though, conclude that the clinical significance of their findings is unknown. Moreover, the sample size was small (14 patients), and deep NMB was not compared with moderate NMB, which is the standard daily clinical practice in gynecologic laparoscopic surgery but with no NMB. It is questionable whether a moderate block was used if the findings were the same. In a similar study, Lindekaer et al. measured the distance from the sacral promontory to the skin during deep NMB and without NMB, in patients undergoing laparoscopic hysterectomy [13]. They concluded that deep NMB increases the distance in a significant manner by 1.57 cm. The sample size was again small (15 patients), and the comparison was not deep NMB with moderate ΝΜΒ but with no NMB.

In another study, Dubois et al. compared deep versus shallow NMB, in patients undergoing laparoscopic hysterectomy, and concluded that permanent optimal surgical conditions were obtained when deep NMB was maintained throughout the operation [11]. The authors report that 14 of 50 patients in the shallow group had instances where operating conditions were deemed unacceptable. However, it should be noted that in 50% of these occurrences, the TOF ratio was above 0.40 (TOF count 4). When the TOF count was two or less, all reported surgical field scores were rated as optimal.

The only study examining the effect of the depth of NMB on postoperative pain in gynecologic laparoscopic surgery is the study of Madsen et al. [14]. Patients were randomized into two groups: deep NMB and low-pressure pneumoperitoneum (8 mmHg) versus moderate NMB and standard-pressure pneumoperitoneum (12 mmHg). The authors concluded that deep NMB and low-pressure might reduce shoulder pain incidence after laparoscopic hysterectomy compared to moderate NMB and standard pressure. Although these results align with our study’s findings, it is unknown which of the two interventions the pain reduction is attributed to.

The most recent study in the field is the study of Soltesz et al. The abdominal wall distension and the surgical conditions of 60 patients scheduled for gynecologic laparoscopic surgery in different degrees of NMB were evaluated. The study failed to reveal any positive effect of intense NMB regarding abdominal wall length compared with moderate or no NMB. Moreover, the existing advantage of intense NMB over no NMB regarding surgical conditions rating (4.7/5 vs. 4.5/5, *p* = 0.025) seems to be of minor clinical relevance [15].

Data from laparoscopic bariatric surgery follow our results regarding the reduction in pain in patients receiving deep NMB. Torensma et al. found that patients on the deep NMB had lower pain scores in the Post-Anesthesia Care Unit (PACU) compared with those in the moderate group (3.9 vs. 4.4, *p* = 0.03) and lower shoulder pain on the ward (1.3 vs. 1.8, *p* = 0.03) [16]. Moreover, regarding patients undergoing laparoscopic gastric bypass, deep NMB markedly improved the surgical conditions compared with moderate block [17]. Finally, the latest studies in colorectal surgery suggest that deep NMB may improve postoperative pain and reduce opioid consumption, findings that agree with our results [18,19].

Our study’s interesting finding is the lower rate of subcutaneous emphysema occurrence in patients receiving deep NMB that reaches statistical significance in operations with a duration of more than 90 min. Subcutaneous emphysema is a known complication of laparoscopic surgery, with the incidence estimated at approximately 0.3% to 3.9% [20]. In our study, the incidence is 12% (17/144), the same as the Lee et al. study (13.5%), which examined the occurrence of this complication in gynecologic laparoscopic surgery [21]. A possible explanation of the higher rates in these two studies is the prospective evaluation and lack of medical record data extraction, where the occurrence of subcutaneous emphysema could have been omitted. Kim et al., in their study on laparoscopic colorectal surgery, found that the patients in the deep NMB group had lower, although non-significant, rates of subcutaneous emphysema (0/30) compared to patients in the moderate NMB group (2/31) [22].

This study’s main strengths are related to its design as a randomized controlled study, in which the blinding of the surgeon was accurate. Moreover, it compares deep versus moderate NMB, which is, in our opinion, the right question regarding laparoscopic surgery, as it is pointless to compare clinical conditions for laparoscopy during deep NMB to no block [23]. The majority of existing evidence, presenting the possible benefits of deep NMB, comes from studies on cholecystectomy or retroperitoneal operations (i.e., prostatectomy and nephrectomy). Data for gynecologic pelvic surgery is scarce, and our study could fill this gap in the literature. The fact that two skilled pelvic surgeons performed all procedures is both a strength and a weakness. It may contribute to obtaining more consistent results in primary outcome evaluation but, at the same time, may not be representative of the median surgical level of physicians performing laparoscopy.

Our study has some limitations. The four-grade rating scale used to evaluate the surgical field is characterized by a high grade of subjectivity; however, it has been used in the past by other, similar studies [11,24,25]. Moreover, we chose to compare analgesic consumption between the two groups to have a more objective evaluation, regarding postoperative pain outcomes, except for patients’ rating. Another limitation of the study is that the mean BMI of our patients (24 kg/m^2^) is within the normal range. There were not enough patients in our sample being overweight or obese to make a subgroup analysis. It would be interesting to see if deep NMB could have any benefit on these patients.

## 5. Conclusions

In conclusion, our data show that deep NMB cannot improve surgical field quality during gynecologic laparoscopic surgery. In contrast, this practice seems to be superior to moderate block regarding postoperative pain in the same type of operations. Deep block reduced the occurrence of subcutaneous emphysema, especially when the operation lasted more than 90 min, but this finding should be interpreted with caution, as it was not a primary endpoint of the study.

## Figures and Tables

**Figure 1 jpm-12-00561-f001:**
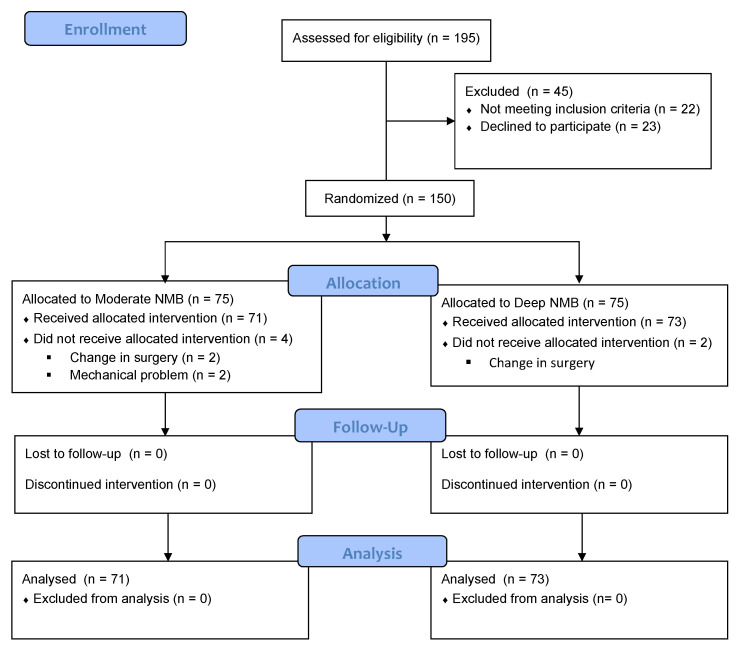
CONSORT flow chart.

**Table 1 jpm-12-00561-t001:** Patient demographic characteristics and surgical data.

	Total (*n* = 144)	Moderateς ΝΜA, (*n* = 71)	Deep ΝΜA (*n* = 73)	*p*
Age, mean ± SD year	37.57 ± 11.14)	36.21 ± 11.32	38.89 ± 10,89	0.14 *
BMI, mean ± SD (Kg/m2)	23.75 ± 3.77	23.56 ± 3.85	23.93 ± 3.71	0.56 *
Smoking	29 (20)	17 (23.9)	12 (16.4)	0.26 ^†^
Drug therapy	35 (24,3)	13 (18.3)	22 (30.1)	0.098 ^†^
Prior abdominal surgery	45 (31,5)	28 (39.4)	17 (23.6)	0.04 ^†^
Laparoscopic surgery				0.23 ^†^
Ovarian cystectomy	58 (40.3)	29 (40.8)	29 (39.7)	
Myomectomy	39 (27.1)	21 (29.6)	18 (24.7)	
Hysterectomy	26 (18.1)	8 (11.1)	18 (24.7)	
Salpingoophorectomy	5 (2.1)	3 (4.2)	2 (2.7)	
Bilateral ovarian cystectomy	10 (6.9)	5 ((7)	5 (6.8)	
Deep endometriosis node dissection	6 (4)	5 (7)	1 (1.4)	
Duration of the operation, mean (range) min	104.34 (60–385)	107.63 (60–325)	101.14 (62–208)	0.95 *
Gas volume Lt, mean (SD)	541 ± 336	559 ± 385	523 ±279	0.96 *
Hemoglobin difference, mean (SD) g/dL	0.96 ± 0.74	0.98 ± 0.79	0.94 ±0.68	0.74 *
Hospital stay, days	2.05 ± 0.88	2.01 ± 0.96	2.08 ±0.80	0.69 *

BMI (Body mass index); data are mean, *n* (%) unless otherwise reported; * analysis with Kruskal–Wallis test; ^†^ analysis with Pearson Chi-square test.

**Table 2 jpm-12-00561-t002:** Surgeon rating of operating field.

	Total (n = 144)	Moderate NMB (n = 71)	Deep NMB (n = 73)	*p*
Small bowel	2.72 ± 0.6	2.63 ± 0.70	2.79 ± 0.47	0.17
Large bowel	2.38 ± 0.78	2.25 ± 0.84	2.51 ± 0.70	0.06
Visualization of operating field	2.56 ± 0.69	2.44 ± 0.77	2.68 ± 0.55	0.06

Data are mean ± standard deviation; Kruskal–Wallis test with Dunnett T3 post hoc test for unequal variances.

**Table 3 jpm-12-00561-t003:** Subgroup analysis of surgeons rating on operating field regarding to operations duration.

Duration ≤ 90 min	Moderate NMB (n = 36)	Deep NMB (n = 33)	*p*
Small bowel	2.58 ± 0.73	2.82 ± 0.39	0.19
Large bowel	2.17 ± 0.78	2.42 ± 0.75	0.12
Visualization of operating field	2.42 ± 0.77	2.67 ± 0.54	0.18
**Duration >90 min**	**Moderate NMB (n = 35)**	**Deep NMB (n = 40)**	** *p* **
Small bowel	2.69 ± 0.68	2.78 ± 0.53	0.57
Large bowel	2.34 ± 0.90	2.58 ± 0.68	0.29
Visualization of operating field	2.46 ± 0.82	2.70 ± 0.56	0.20

Data are mean ± standard deviation; Kruskal–Wallis test with Dunnett T3 post hoc test for unequal variances.

**Table 4 jpm-12-00561-t004:** Patient rating of postoperative symptoms and postoperative bowel function.

	Total (n = 144)	Moderate NMB (n = 71)	Deep NMB (n = 73)	*p*
Postoperative pain *	1.58 ± 1.24	2.39 ± 1.07	0.79 ± 0.80	<0.001
Postoperative bloating *	0.95 ± 0.89	1.11 ± 0.95	0.79 ± 0.82	0.033
Postoperative nausea *	0.90 ± 1.15	1.13 ± 1.24	0.67 ± 1.02	0.017
Day of first passage of gases ^†^(days)	082 ± 0.59	0.79 ± 0.58	0.85 ± 0.59	0.538
Day of first passage of stool ^†^ (days)	1.81 ± 0.70	1.86 ± 0.72	1.75 ± 0.68	0.368

Data are mean ± standard deviation; * 5-point Likert scale; ^†^ day of incidence postoperatively; parametric analysis of variants with one-way ANOVA.

**Table 5 jpm-12-00561-t005:** Postoperative analgesic use.

Analgesic Consumption	Moderate NMB (n = 71)	Deep NMB (n = 73)	*p*
Typical analgesia	58 (81.7)	70 (95.9)	0.007
Εxtra opioid analgesia	13 (18.3)	3 (4.1)	
Date are n (%)			
Pearson Chi-square test			

Data are n (%); Pearson Chi-square test.

**Table 6 jpm-12-00561-t006:** Overall incidents of subcutaneous emphysema and subgroup analysis regarding operation duration.

	Moderate NMB (n = 71)	Deep NMB (*n* = 73)	*p*
Subcutaneous emphysema	12 (16.9)	5 (6.8)	0.06
	**Moderate NMB (*n* = 36)**	**Deep NMB (*n* = 33)**	
Subcutaneous emphysema for operations ≤90 min	2 (5.6)	1 (3)	0.6
	**Moderate NMB (*n* = 35)**	**Deep NMB (*n* = 40)**	
Subcutaneous emphysema for operations >90 min	10 (28.6)	4 (10)	0.004

Data are n (%); Pearson Chi-square test.

## Data Availability

Data repository direct link: https://data.mendeley.com/datasets/8sg3yx672r/1. Date of last access: 18 February 2022.

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
