# Peer review of "Deep versus Moderate Neuromuscular Blockade in Gynecologic Laparoscopic Operations: Randomized Controlled Trial"

_jpm, 2022, doi:10.3390/jpm12040561_

Round 1

Reviewer 1 Report

The manuscript presented by the authors investigates if deep neuromuscular blockage (NMB) improves surgical conditions and post-operative pain compared to moderate block in patients undergoing gynecologic laparoscopic surgery. There are a few shortcomings that need to be addressed before considering the manuscript for publication.

Comment:

  • The abstract should be standalone and acronyms like NMB are not explained where they first appear in the abstract.
  • Introduction, line 55 “So far, data regarding the effect of deep NMB on gynecologic laparoscopy are scarce…” This seems to be false. Kindly find the following links to studies in this area. Authors should specify how their study is unique or different.
    1. https://www.ncbi.nlm.nih.gov/pmc/articles/PMC6056743/
    2. https://asja.springeropen.com/articles/10.1186/s42077-020-00073-y
    3. https://www.intechopen.com/chapters/52107
    4. https://www.sciencedirect.com/science/article/pii/S0007091217300478
    5. https://www.mdpi.com/2077-0383/9/4/1078/htm
  • Reference: The literature review is 2018 and older, no new reports are included in the manuscript. The authors should update it.

Author Response

Response to the reviewer:

Point 1: The abstract should be standalone and acronyms like NMB are not explained where they first appear in the abstract.

Response to point 1: Thanks a lot for the comment. We inserted an explanation for NMB the first time it appears in the Abstract.

Point 2: Introduction, line 55 “So far, data regarding the effect of deep NMB on gynecologic laparoscopy are scarce…” This seems to be false. Kindly find the following links to studies in this area. Authors should specify how their study is unique or different.

Response to point 2: Authors want to thank the reviewer for the comment. We agree with the reviewer and we have changed the false expression in the manuscript. We also found the suggested studies useful and we inserted in the references and added a few comments in the main text. Finally, we tried to clarify the scope of the study in the introduction as suggested.

Point 3: Reference: The literature review is 2018 and older, no new reports are included in the manuscript. The authors should update it.

Response to point 3: This comment helped the authors to strengthen the manuscript as we included latest reposrts in the field in the references and in the main text as well.

Reviewer 2 Report

Dear Sir/Madame

I have review carefully the article. The subject is indeed less studied. In my opinion the major lack of the current study is its heterogeneity: the inclusion of a large variety of laparoscopic gynecological procedures from ovarian cystectomy to hysterectomy. It is true that the analysis was compared for each type of procedure in comparison by both groups. 

I think the article worth to be published. I encourage you to continue your work and to study the impact of these two type of anesthesia on a single procedure such as hysterectomy.

Author Response

Point 1: I have review carefully the article. The subject is indeed less studied. In my opinion the major lack of the current study is its heterogeneity: the inclusion of a large variety of laparoscopic gynecological procedures from ovarian cystectomy to hysterectomy. It is true that the analysis was compared for each type of procedure in comparison by both groups.

Response to point 1: Authors are excited that the reviewer finds their work interesting and definitely agree with the reviewer regarding the literature gap in the field. We recognise the heterogeneity regarding the variety of the laparoscopic procedures included in the study. On the other hand, we tried to include the most common laparoscopic surgeries performed in a gynaecology endoscopic unit aiming to make our findings more generalisable for the majority of the gynecologic departments. To partially  overcome the heterogeneity issue we tried to divide the operations according to their duration, in less than 90min and more than 90min and proceed to subgroup analysis. This way we tried to achieve more consistent and reliable results. 

Point 2: I think the article worth to be published. I encourage you to continue your work and to study the impact of these two type of anaesthesia on a single procedure such as hysterectomy.

Response to point 2: Authors want to thank the reviewer that he judges their work "worth to be published". The design of a more focused study to include laparoscopic hysterectomy is in our future plans.